# Extracorporeal Membrane Oxygenation (ECMO)-Associated Coagulopathy in Adults

**DOI:** 10.3390/diagnostics13233496

**Published:** 2023-11-21

**Authors:** Frantzeska Frantzeskaki, Dimitrios Konstantonis, Michail Rizos, Vasileios Kitsinelis, Georgios Skyllas, Ioannis Renieris, Maria Doumani, Vasileios Kolias, Eirini Kefalidi, Dimitrios Angouras, Argyrios Tsantes, Iraklis Tsangaris

**Affiliations:** 12nd Critical Care Department, Attikon University Hospital, Medical School, National and Kapodistrian University of Athens, 157 72 Athens, Greece; d.konstantonis@gmail.com (D.K.); rizosm1965@gmail.com (M.R.); v.kitsinelis@gmail.com (V.K.); georgskyllas@gmail.com (G.S.); ioannisrenieris3@gmail.com (I.R.); doumani91@hotmail.com (M.D.); itsagkaris@med.uoa.gr (I.T.); 2Department of Cardiac Surgery, Attikon University Hospital, Medical School, National and Kapodistrian University of Athens, 157 72 Athens, Greece; vasilioskollias@gmail.com (V.K.); eirkef.ek@gmail.com (E.K.); dangouras@yahoo.com (D.A.); 3Laboratory of Haematology, Blood Bank Unit, Attikon University Hospital, Medical School, National and Kapodistrian University of Athens, 157 72 Athens, Greece; atsantes@yahoo.com

**Keywords:** extracorporeal membrane oxygenation, coagulation, immunothrombosis, viscoelastic tests

## Abstract

Extracorporeal membrane oxygenation (ECMO) is used for the management of severe respiratory and cardiac failure and as a bridge to achieve definite treatment or transplantation. ECMO-associated coagulopathy (EAC) is a frequent complication leading to high rates of thrombosis or severe haemorrhage, contributing to morbidity and mortality among patients. Understanding the pathophysiology of EAC is substantial for effectively managing patients on ECMO. We analyse the underlying mechanism of EAC and discuss the monitoring of the coagulation profile, combining the viscoelastic point-of-care assays with the conventional coagulation laboratory tests.

## 1. Introduction

Extracorporeal membrane oxygenation is used in ICU for the management of severe respiratory and cardiac failure. Despite recent advances in circuit and membrane technology, the reported morbidity and mortality remain high [1]. Mortality is attributed to unsuccessful cardiopulmonary recovery, while some serious complications are usually encountered. ECMO-associated coagulopathy (EAC) is a frequent complication leading to high rates of thrombosis or severe haemorrhage [2]. Interestingly, bleeding complications are strongly associated with a poor outcome and mortality [3,4]. The elucidation and monitoring of the underlying factors of EAC substantially contribute to effectively managing patients on ECMO. The aim of the current article is to analyse the pathomechanism of EAC to highlight the coagulation monitoring requirements and to conclude with future recommendations.

## 2. Interactions between Coagulation Cascade and Biosurfaces Leading to EAC

EAC might be associated with clinical thrombosis and haemorrhage, reported in 15–85% and 15–21% of patients, respectively [5]. In a recent analysis of the Extracorporeal Life Support Organization (ELSO) registry, haemorrhage was observed in 37% of patients, while 21% of patients experienced both bleeding and thrombotic events [6]. The thrombosis of the oxygenator membrane is observed in 10–15% of ECMO patients [7]. The prothrombotic effect of the interaction between blood and biosurfaces is described by the three components of Virchov’s triad: blood exposure to the artificial endothelium consisting of the biosurface of the oxygenator membrane; the abnormal blood flow through the ECMO circuit; and the hypercoagulability caused by the release of free hemoglobulin into circulation during the haemolysis effect [5]. The prevention of thrombosis during ECMO is commonly achieved by the continuous infusion of unfractionated heparin, although there is no consensus on the appropriate anticoagulation strategy.

At the same time, haemorrhagic complications might be caused by the administration of unfractionated heparin, thrombocytopenia, platelet dysfunction, acquired von Willebrand syndrome and the consumption of coagulation factors [8]. According to a review of the ELSO registry, the risk factors associated with haemorrhage in patients receiving veno-venous (VV) ECMO support for respiratory failure are cardiac arrest, precannulation hypoxemia, metabolic acidosis, and shock [9]. A prediction model including these risk factors was successfully used in order to identify ECMO patients with an increased probability for haemorrhage. ECMO circuit thrombosis might be accompanied by haemorrhage, rendering patients’ subsequent management rather challenging. The transfusion of blood products, the administration of coagulation factors, or fibrinolysis inhibitors might aggravate ECMO circuit failure. The subsequent formation of thrombi in the ECMO circuit will deteriorate the coagulopathy, leading to a vicious cycle which might require the exchange of the circuit and oxygenator [10].

Moreover, the patient’s underlying disease might be associated with prothrombotic or bleeding effects in the context of trauma-induced coagulopathy, disseminated intravascular coagulation (DIC), or COVID-19-associated coagulopathy. The treatment algorithm of COVID-19 patients with ARDS frequently includes ECMO support. The hyperinflammatory state produced by COVID-19 might interact with the effect of the exposure of blood to biosurfaces, producing bleeding or thrombotic complications [11]. Indeed, according to several reports, the bleeding rate in COVID-19 patients managed with ECMO is 14–42%, while the reported intracranial haemorrhage rate is about 4% [12,13]. Interestingly, this rate is higher than the respective central nervous system bleeding rate (2%) reported in EOLIA (ECMO to rescue lung injury in severe ARDS) study [14]. COVID-19-associated endotheliopathy and higher doses of anticoagulants might explain the above difference. Moreover, rates of thrombosis are common in hospitalised COVID-19 patients, ranging between 20 and 30%, while pulmonary embolism is the most common thrombotic manifestation [15,16]. Thrombotic complications are commonly reported in COVID-19 patients managed with ECMO support, despite the higher dose of anticoagulant regimens compared to non-COVID-19 ECMO patients [6]. 

EAC is initially triggered by the exposure of blood to extracorporeal surfaces, which consequently leads to the activation of coagulation factors, the complement system, platelets, von Willebrand factor (VWf), and fibrinolysis (Figure 1). Haemodilution caused by the priming volume of the ECMO circuit results in a decrease in haematocrit and coagulation factors [17]. Moreover, the interaction between blood and the artificial surfaces of the circuit leads to the absorption of fibrinogen and proteins by the surfaces, known as a “Vroman effect” [18]. Coagulation factors, such as a high molecular weight kininogen (HMWK), factor XII, albumin, immunoglobulins, and complement components (C3) are absorbed by the fibrinogen layer whilst it undergoes modification, which renders their molecules thrombogenic [19]. The protein and fibrinogen layer which are formed on the artificial “endothelium” of the ECMO circuit trigger the activation of coagulation, which in turn regulates the inflammatory process induced by the innate host response. The major pathological role of coagulation in the innate host response has recently been elucidated and defined as immunothrombosis, a phenomenon playing an important role in the pathophysiology of sepsis and ARDS [20]. Therefore, platelets accumulate at the site of the artificial “endothelium” at the biosurface. Their activation leads to the release of the content of their granules, consisting in immunomodulatory mediators, antimicrobial peptides (AMPs), and microparticles (MPs). These molecules contribute to the migration of neutrophils and the activation of the complement. The platelet–neutrophil complex shows different properties than platelets or neutrophils alone in terms of increased adhesion and molecule expression, greater phagocytotic activity, the production of toxic oxygen radicals, and the expression of pro-inflammatory cytokines, such as interleukins–1β, -6, and -8, as well as tumour necrosis factor (TNF-α), which amplifies the inflammatory response and generates the immunothrombotic procedure (Figure 1) [21]. Indeed, the levels of proinflammatory cytokines rapidly increase after the initiation of ECMO circulation, contributing to endothelial damage [22].

Platelets’ adhesion to the artificial “endothelium” of the surfaces of the ECMO circuit plays an orchestral role in the immunothrombosis procedure. Severe thrombocytopenia (<50 × 10^9^/L) is common in patients on ECMO, requiring frequent transfusions, independently of the duration of the extracorporeal circulation [23]. Additionally, shear stress caused by blood passing through the ECMO circuit, results on impaired platelets’ aggregation and the loss of VWF multimers [24]. This phenomenon has been described 15 min after extracorporeal circuit initiation and can last during the whole procedure. Reduced glycoprotein (GP) Ibα and GPVI levels, which are receptors for VWF and collagen, respectively, are implicated in the platelet dysfunction [24]. Platelet dysfunction is confirmed by the lower aggregation demonstrated by light transmission aggregometry, and consequently results in haemorrhagic complications [25]. Additionally, microthrombi might be produced within the surfaces of the ECMO circuit, and subsequently be embolised in vital organs. Neurologic complications have been reported in patients during extracorporeal circulation [26].

Platelet-derived microparticles (MPs) might also be implicated in EAC. MPs are shed from precursor cells after several triggering factors such as inflammation or shear stress caused by extracorporeal blood circulation [27]. They preserve the lipid bilayers of parent cells, while they contain proteins, ribosomal RNA, messenger RNA, and microRNA [28]. These mediators are secreted by MPs towards various target cells, generating intercellular information exchange. MPs’ membranes contain large quantities of phosphatidyloserine, providing a suitable environment for the activation of the coagulation cascade [29].

TF plays a constitutional role in the activation of coagulation, which is tightly correlated with the inflammatory process [30,31]. It is a membrane protein expressed in the fibroblasts of the adventitia of vessels, but it is also distributed by epithelial cells, endothelial cells, platelets, and monocytes, particularly under inflammatory conditions. According to the cell-based model of coagulation, TF is released into the bloodstream and interacts with the proteases of the coagulation cascade. Therefore, TF binds to Factor VII, while the TF: Factor VIIa complex consequently activates Factor X. Factor Xa and Factor Va constitute the prothrombinase complex in the presence of calcium and the phospholipid surface provided by activated platelets. The prothrombinase complex activates prothrombin to thrombin. Thrombin leads to the ample formation of fibrin and microthrombi. There are several reports showing an increased expression of TF in the monocytes in Platelets’ dysfunction the biosurfaces of ECMO circuits in vitro, resulting in a procoagulant effect [31,32].

Contact pathway activation might also contribute to EAC, as it has been shown in neonates on ECMO in whom the kallikrein inhibitory capacity was decreased, while thrombin–antithrombin complexes were present [33]. The reduced levels of FXII and increased levels of FXIIa and precallikreine were demonstrated in patients on ECMO [34,35]. Other coagulation factors might be compromised during ECMO, contributing to bleeding or thrombotic diathesis: the level of FVIII was decreased in the animal models on ECMO [36], in parallel with decreases in fibrinogen and VWF. Decreased thrombin generation associated with decreased levels of factor X and prothrombin has been described in patients on extracorporeal circulation [37], while in ECMO patients with severe haemorrhagic complications, decreased levels of factors VII and X were measured [38].

Acquired von Willebrand disease is also implicated in the pathophysiology of EAC. The high-molecular-weight polymers of VWF are cleaved by metalloproteinases, due to the shear stress caused by ECMO circulation. The decreased levels of large polymers of vWF have been demonstrated in patients with veno-arterial or veno-venous ECMO within one day of extracorporeal circulation [39]. Therefore, platelet adhesion, a process promoted by vWF, is compromised, resulting in bleeding complications, especially from the respiratory system, puncture sites, and mucosa [40]. vWF levels normalise in 24 h after the end of ECMO treatment [25].

The activation of the complement system contributes to the development of EAC. The complement pathway is activated after the binding of immunoglobulin on the artificial “endothelium” of ECMO circuit. The excessive complement cascade activation cannot be supressed, since the normal endothelial regulatory mechanisms are absent, which causes capillary leak syndrome and inflammatory response [6,38]. Complement degradation leads to the production of anaphylatoxins, which contribute to the subsequent inflammatory reaction. Moreover, anaphylatoxins might bind to the ECMO circuit, participating in the immunothrombosis procedure [20]. Complement degradation products contribute to the adhesion of granulocytes and monocytes to ECMO surfaces, as a result of the activation of respectivereceptors [41]. Moreover, anaphylatoxins pass in the systemic circulation, interacting with the proteases of the coagulation cascade, platelets, and endothelial cells. Consequently, the activation of platelets and endothelial cells increased the expression of TF and further vWF production, playing an important role in coagulation disorders [42,43,44].

The derangement of fibrinolysis might also be involved in EAC despite the lack of sufficient data. Thrombin generation and several other situations as hypoxia, trauma, or sepsis trigger the release of the tissue plasminogen activator (t-PA), which activates plasminogen and plasmin, leading to fibrin degradation. In ECMO patients, a decrease in t-PA and plasminogen activator inhibitor (PAI-1) levels is observed, followed by a respective increase in both [41]. Hyperfibrinolysis and hypofibrinaemia has been observed in patients on extracorporeal circulation, while the administration of antifibrinolytic agents might reduce the subsequent haemorrhagic complications [45]. According to a recent prospective study in a cohort of 30 patients on ECMO support, endogenous fibrinolysis might result in haemorrhagic complications. Indeed, increasing D-Dimers are a surrogate marker for the development of fibrinolytic activity [46]. Moreover, hypofibrinogenaemia and increased levels of fibrin degradation products in ECMO patients might be indicative of oxygenator-induced hyperfibrinolysis, requiring circuit exchange.

Finally, haemolysis is also implicated in EAC. Increased levels of free plasma hemoglobulin have been measured in 67% of ECMO patients, while levels higher than 50 mg/dL are associated with increased mortality [44]. The presence of free hemoglobulin contributes to the prothrombotic state by binding nitric oxide (NO). Decreased levels of NO result in vasoconstriction and further platelet activation and adhesion [42]. Moreover, haemolysis leads to the release of red blood cell-derived microparticles which also exert prothrombotic effects [43].

## 3. Monitoring Coagulation Profile of ECMO Patients

There are no current guidelines defining the optimal method of monitoring the overall haemostatic profiles of patients on extracorporeal circulation. Moreover, since the monitoring of heparin effect is performed in vitro, it does not take into consideration the in vivo interaction between the endothelium, coagulation cascade, and oxygen membrane that leads to EAC. The methods used for assessing EAC and the heparin effect are plasma-based or whole blood-based. Plasma-based tests do not take into consideration platelets’ dysfunction or the impairment of clot formation. Viscoelastic tests such as thromboelastography and thromboelastometry are whole blood tests that dynamically estimate clot formation, although not broadly available and not easily interpreted. The advantages and disadvantages of monitoring technics for EAC and anticoagulation therapy are depicted in Table 1. 

## 4. Activated Clotting Time (ACT)

ACT is a whole-blood, point-of-care assay, traditionally used for the assessment of the anticoagulant effect of heparin in patients on the cardiopulmonary bypass [47]. However, many confounding factors, especially in the critical care setting, might affect the accuracy of ACT as an assay for monitoring the heparin effect on ECMO patients. These include hypothermia, inflammation, liver failure, and thrombocytopenia [48]. In a recent retrospective analysis of 604 children on ECMO, ACT values did not show a significant correlation with the heparin dose [49]. Therefore, monitoring the heparin effectiveness by ACT might lead to suboptimal coagulation assessment and the subsequent adjustment of anticoagulants dose in critically ill patients.

## 5. Activated Partial Thromboplastin Time (aPTT) and Anti-Factor Xa Level

aPTT is a plasma-based assay, measured by adding calcium to plasma after exposure to a contact activator [50]. There are no randomised controlled studies on the target levels of aPTT in ECMO patients, despite the fact that aPTT, which is 1.5–2.5 times patient’s baseline, has historically been correlated with a decreased risk of thromboembolism [51]. Therefore, the therapeutic aPTT range is set to 1.5–2.5 times patient’s baseline aPTT, although this was not validated for ECMO patients in randomised controlled studies. Moreover, in critically ill patients, acute phase proteins and elevated fibrinogen might diminish the aPTT value, reflecting coagulation abnormalities rather than the clear effect of heparin. A retrospective study in 149 patients on ECMO demonstrated the significant correlation between higher aPTT values and the risk of bleeding [52]. Given the variability of aPTT measurements, monitoring the anticoagulant effect of heparin is often performed by the anti-factor Xa assay. 

Anti-Xa is a plasma-based test that measures the heparin effect by evaluating the ability of heparin to catalyse antithrombin’s inhibition of the factor Xa activity. Therefore, anti-Xa directly measures the heparin effect without evaluating the overall coagulation profile of ECMO patients, in the pathophysiology of which several other procoagulant factors are implicated. Thus, since aPTT is affected by the heparin dose and other coagulation factors, it may better predict the bleeding risk due to EAC as compared to anti-Xa. This was well indicated in a retrospective study including 34 ECMO patients, in whom anti-Xa and/or aPTT was measured. High aPTT values were predictive of bleeding, while low anti-Xa values were associated with thrombosis [53]. Moreover, a better correlation between the heparin dose and anti-Xa was observed, as compared to aPTT. Therefore, an accurate evaluation of the bleeding and thrombotic risk should be based on a combination of the aPTT and anti-Xa measurement. Moreover, anti-Xa levels are not influenced by the presence of the lupus anticoagulant, liver disease, and consumptive coagulation disorders [54].

## 6. Viscoelastic Haemostatic Assays

Thromboelastography (TEG) and thromboelastometry (ROTEM, ClotPro^®^) are whole-blood, point-of-care assays providing a global assessment of the viscoelastic properties of the clot, including clot initiation (evaluating coagulation factors and the heparin effect), amplitude (evaluating platelets and fibrinogen), and stability (fibrinolysis). Multiple assays are available for each device to evaluate the extrinsic and intrinsic coagulation pathway, the contribution of fibrinogen in clot formation, and heparinase assays to the inhibit heparin effect. ROTEM and TEG are recommended in surgical or trauma patients for the achievement of goal-directed transfusion therapy during bleeding [55]. However, there are less data regarding the use of viscoelastic assays in ECMO patients. According to recent studies, TEG or ROTEM showed hypercoagulability and might predict thrombotic complications [50], while a reduced maximum clot firmness (MCF) has been associated with bleeding [56]. Moreover, in case studies, ROTEM has been used for the differential diagnosis of EAC and the documentation of hyperfibrinolysis, a phenomenon that might impose ECMO circuit exchange [57]. In a study including 57 patients, it was shown that V-V ECMO therapy led to a progressive increase in the clotting time (CT) and a decrease in MCF, while ROTEM assays did not help the prediction of haemorrhage [58]. A randomised controlled study comparing a TEG-based protocol to an aPTT-based one to assess the heparin anticoagulation demonstrated that TEG can be safely used for monitoring coagulation on ECMO patients [59]. ClotPro^®^ (Santair Medial Applications, Athens, Greece) has also been used for the assessment of EAC in COVID-19 patients on ECMO support [60]. More studies are required to determine the exact role of the viscoelastic assays in the monitoring of EAC and in consequent therapeutic interventions.

## 7. D-Dimers, Platelets, and Antithrombin

Fibrin degradation products, including D-Dimers, are formed after fibrin formation and degradation. Although D-Dimers’ specificity is low, daily measurements are recommended in ECMO patients for the early diagnosis of the thrombosis of the membrane oxygenator. Low fibrinogen levels, rising D-Dimers, and elevated free hemoglobulin levels might indicate the possible thrombosis of the circuit, requiring prompt oxygenator change. D-Dimers are also a surrogate marker of excessive fibrinolytic activity, which results in haemorrhagic complications during ECMO support [46]. 

Platelets should also be measured on a daily routine, as gradual thrombocytopenia might become heparin-induced thrombocytopenia (HIT). HIT has been described in 4–7% of ECMO patients [61]. In COVID-19 patients under ECMO support, the reported incidence of HIT is higher (10.52%) contributing to thrombotic complications [62]. In the case of suspected HIT, heparin administration should be stopped. A non-heparin anticoagulant should be administered and an enzyme-linked immunosorbent assay for antiplatelet factor 4 antibodies may be performed to confirm the HIT. 

Light-transmission aggregometry (LTA) is the gold standard assay for the assessment of platelet aggregation. Multiple-electrode aggregometry (MEA) has been used as a point-of-care technique for the evaluation of platelet aggregation in patients undergoing a cardiopulmonary bypass [63]. Platelet aggregation is usually impaired in patients on ECMO, as has been shown in the experimental models of an artificial circuit and in small studies in humans [64]. However, ECMO-related thrombocytopenia might influence the accurate assessment of platelet function. Moreover, haemolysis might also compound platelet aggregation evaluation [65]. Bleeding complications in ECMO patients might be caused by impaired platelet aggregation, as confirmed by LTA [66]. Despite the normal count of platelets, transfusions and the administration of tranexamic acid should be considered. 

Antithrombin (AT) deficiency may result in “heparin resistance”, as AT is required as a cofactor for heparin to achieve the antithrombotic effect. The role of AT in the coagulation procedure consists in the inhibition of thrombin and the activation of factor X after forming a complex with thrombin. Interestingly, AT administration to patients on V-V ECMO was associated with a more rapid decrease in pro-inflammatory cytokines compared to the placebo, demonstrating the anti-inflammatory effect of AT, which might be independent of its anticoagulative results [67]. The anti-inflammatory effect of AT is contributed to prostacyclin’s release, to the reduction in the activity of P-selectin, and to the prevention of the activation of leukocytes, resulting in a decrease in pro-inflammatory cytokines. However, there is not adequate evidence for recommending routine AT monitoring and the subsequent supplementation in patients on ECMO. 

## 8. Differences between Veno-Venous (V-V ECMO) and Veno-Arterial (V-A ECMO)-Associated Coagulopathy

Several studies have demonstrated significant differences between V-V and V-A ECMO patients in terms of coagulation parameters and platelets. Fibrinogen plasma levels and fibrin polymerisation are only increased in patients under V-V ECMO support, leading to a prothrombotic state [39]. This phenomenon can be explained by the pathophysiology of ARDS, leading to refractory hypoxemia and requiring V-V ECMO support. A key characteristic of ARDS is an imbalance between inflammation and coagulation, resulting in the complex procedure of immunothrombosis, which consequently affects the respective coagulation parameters [20]. Moreover, in V-V ECMO patients, GP IIb/IIIa activation is increased, compared to V-A ECMO patients, contributing to platelet activation, and consequently, to the immunothrombosis process.

Despite the subsequent prothrombotic state, V-V ECMO is associated with an increased intracranial haemorrhage incidence compared to V-A ECMO, as it has been demonstrated in ELSO studies [68]. According to other reports, although systemic thromboembolism and bleeding in V-A ECMO patients might result in a higher probability of intracranial haemorrhage, no significant difference was demonstrated between V-V and V-A ECMO support [69]. Arterial cannulation in the V-A ECMO support might lead to the earlier activation of the coagulation cascade and subsequent coagulopathy. A recent prospective analysis of a cohort of 30 patients on V-V or V-A ECMO demonstrated that bleeding complications were more common in patients under V-A ECMO [46]. Further studies are required in order to investigate the association between ECMO modality with coagulopathy. Moreover, there are not sufficient data regarding the differences in coagulopathy between central and peripheral ECMO.

## 9. Recommendations

There is a paucity of evidence on optimal anticoagulation management in ECMO patients. Current recommendations are based on retrospective studies, demonstrating the safety of conservative anticoagulation strategies or even no anticoagulation, especially in V-V ECMO patients [70]. Several studies have shown that heparin-based anticoagulation targeting over 1.5 times patients’ baseline aPTT resulted in more bleeding events without reducing mortality or thrombotic complications [71,72]. According to a recent randomised study, a low heparin dose (aiming at aPTT < 45 s) compared to the usual dose (aPTT 50–70 s) was associated with a significant decrease in aPTT and anti-Xa, while the rate of bleeding or thrombotic complications remained the same [73]. A recent retrospective analysis of all ECMO patients in a single centre, during eight years, showed that a high level of coagulation (aPTT ≥ 55 s) was significantly associated with fatal bleeding events without reducing the rate of thrombotic complications [74]. A meta-analysis of seven studies comparing the safety and effectiveness of the low-dose anticoagulation strategy to a standard dose in ECMO patients confirmed that the low dose was associated with less haemorrhagic gastrointestinal and surgical site complications, while the mortality and the rate of thrombotic events and intracranial haemorrhage was similar between the two groups of patients [75].

Another recent meta-analysis of 17 studies regarding the use of aPTT for anticoagulation monitoring in ECMO patients demonstrated that bleeding complications were observed in patients with a longer aPTT, longer ECMO support, and higher mortality. However, the study did not confirm any association between the aPTT threshold and risk of haemorrhage [76]. Moreover, the complex mechanism of EAC resulting in impaired haemostasis, renders the concept of ‘anticoagulation-free’ ECMO support feasible and reasonable. Indeed, there are several retrospective studies demonstrating the safety of not using anticoagulation [71]. A recent systemic review on ECMO support without therapeutic anticoagulation demonstrated that the incidence of thrombosis was comparable to ECMO patients who traditionally received systemic anticoagulation while the adverse events profile remained similar [77]. Given the paucity of robust data on the ideal anticoagulation strategy, ongoing studies (RATE, SAFE-ECMO, A FREE ECMO) aim to confer individualised recommendations for adult patients under ECMO support. 

Regarding the transfusion strategy of blood products, there is still a lack of evidence. According to current ELSO recommendations, hemoglobulin concentration should be 14–15 g/dL, while platelets’ transfusion should be considered in order to maintain platelet levels > 100,000 × 10^9^/L in a bleeding patient and >50,000 × 10^9^/L in a non-bleeding patient [50]. However, more conservative transfusion strategies were proposed given the risk of fluid overload, which is strongly associated with mortality. According to a recent survey, an hemoglobulin transfusion threshold of 8–10 g/dL was used in most centres, while the platelets transfusion threshold was 50,000 × 10^9^/L and fibrinogen’s threshold was 2 g/L [78]. Lower thresholds are commonly considered in non-bleeding patients. 

Severe haemolysis is strongly associated with circuit thrombosis, and can be easily recognised by measuring the free hemoglobulin plasma concentration, which is implicated in acute renal failure and mortality [44]. Other markers of haemolysis and the subsequent requirement of circuit exchange are D-Dimers and platelets’ count [79]. Our recommendations for future practice on aPTT and anti-Xa target levels and on transfusion strategies and anticoagulation management are depicted in Table 2.

## 10. Future Perspectives

There are limited data on other point-of-care tests to assess EAC and monitor heparin activity. PFA-100 is quite a sensitive diagnostic test for von Willebrand disease, although it has not been validated in ECMO patients [80]. Moreover, a mass spectrometry method has recently been used for the measurement of vWF cleavage, in patients with left ventricular assist devices [81]. The global thrombosis procedure has also been assessed by a point-of-care test, which identifies the prothrombotic risk caused by impaired fibrinolysis. The test has been used in acute coronary syndromes and might also be useful in ECMO patients, as it evaluates the whole coagulation procedure under conditions of shear stress caused by extracorporeal circulation [82,83]. Finally, the thrombin generation test is based on the stimulation of coagulation by the exogenous tissue factor. The value of the subsequent endogenous thrombin potential (ETP) in ECMO patients might be associated with the probability of venous thrombosis or haemorrhage according to small studies [84,85]. Interestingly, there have been several reports on heparin free V-V ECMO, especially as a bridge to lung transplantation in order to avoid perioperative bleeding caused by coagulopathy [86]. This assessment is based on the administration of AT during the perioperative period, the use of heparinised ECMO circuit, and the control of fibrinolysis with tranexamic acid. More clinical studies are required in order to elucidate the role of these tests in the early identification of bleeding or prothrombotic risk in ECMO patients and in the subsequent management of anticoagulant therapy.

## 11. Conclusions

EAC is a multifactorial syndrome associated with thrombotic or bleeding complications contributing to morbidity and mortality. Despite the evolution of the circuit and oxygenator technology, the extracorporeal circulation of blood is associated with the activation of haemostasis, while the shear stress caused by pumps inevitably leads to the haemolysis and damage of large-protein molecules. The activation of the intrinsic and extrinsic coagulation pathway, immunothrombosis, and haemolysis are associated with circuit thrombosis and systemic thromboembolism. Fibrinolysis, the decrease in fibrinogen, and acquired Von Willebrand syndrome combined with the administration of therapeutic anticoagulation might lead to haemorrhagic complications. Point-of-care viscoelastic analyses including thromboelastometry, as well as conventional coagulations tests, help in the identification of coagulation disorders and subsequent goal-directed transfusion therapy. More studies are required for the further understanding of the pathophysiology of EAC and for the development of treatment algorithms in order to improve ECMO outcomes in patients with refractory hypoxemia or end-stage lung disease.

## Figures and Tables

**Figure 1 diagnostics-13-03496-f001:**
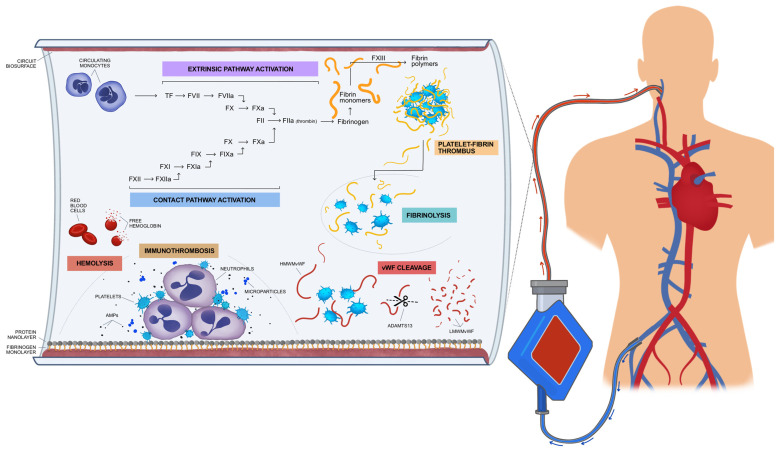
ECMO-associated coagulopathy: The exposure of blood to the artificial endothelium consisted of the biosurface of the oxygenator membrane, consequently leading to the activation of coagulation factors, the complement system, platelets, and von Willebrand factor (VWf) and fibrinolysis. TF is released into the bloodstream and interacts with the proteases of the coagulation cascade, while the prothrombinase complex activates prothrombin to thrombin. Thrombin leads to the ample formation of fibrin and microthrombi. The major pathological role of coagulation in the innate host response is defined as immunothrombosis. Therefore, platelets accumulate at the site of the artificial “endothelium” of the biosurface. Their activation leads to the release of the content of their granules, consisting in immunomodulatory mediators, antimicrobial peptides (AMPs) and microparticles (MPs). These molecules contribute to the migration of neutrophils and the activation of the complement. The derangement of fibrinolysis and acquired von Willebrand disease might also be involved in EAC. High-molecular weight-polymers of VWF are cleaved by metalloproteinases, due to the shear stress caused by ECMO circulation. Haemolysis is also implicated in EAC, as free hemoglobulin contributes to the prothrombotic state.

**Table 1 diagnostics-13-03496-t001:** Tests for coagulation monitoring in ECMO patients.

Test	Advantages	Disadvantages	Normal Range
ACT	- Point of care - Whole blood - Useful in CABG	- Confounding factors: hypothermia, inflammation, liver failure, thrombocytopenia - Suboptimal coagulation assessment in critically ill	70–180 s
aPTT	- Plasma-based - High levels predict bleeding risk	- Confounding factors in critically ill: fibrinogen levels, acute phase proteins, consumption of coagulation factors during bleeding or thrombosis, lupus inhibitor - Variability between laboratories - Time delay	21–35 s
Anti-Xa	- Plasma-based - Direct measurement of heparin effect - Low levels predict thrombosis risk - Not influenced by liver disease, consumptive coagulopathy, lupus anticoagulant	- Expensive - Time delay	<0.1 U/mL
Viscoelastic assays	- Point of care - Whole blood - Information about hypercoagulability, hyperfibrinolysis - Differential diagnosis of EAC	- Expensive - Not widely available - Limited data on ECMO patients	* CT: 38–79 s CFT:34–159 s A10: 43–65 mm MCF: 50–72 mm LI 60: ≥60%

Abbreviations: CABG: coronary artery bypass graft; ACT: activated clotting time; aPTT: activated partial thromboplastin time; EAC: ECMO-associated coagulopathy; CT: clotting time; CFT: clot formation time; MCF: maximum clot firmness; LI 60: lysis index 60 min after CT. * Viscoelastic assays parameters are referred to EXTEM (extrinsic haemostasis rotational thromboelastometry).

**Table 2 diagnostics-13-03496-t002:** Anticoagulation, blood product goals, and management during ECMO.

Test	Goals	Management
aPTT	1.5–2 patient’s baseline	Heparin range change 10–20%
Anti-Xa	0.3–0.7 U/Ml	Heparin range change 10–20%
Hemoglobin	>7–9 g/dL	RBC 10 mL/kg
Fibrinogen	>1.5–2 g/L	Cryoprecipitate 1 unit/5 kg
Platelets	≥100,000 × 10^9^/L in a bleeding patient ≥50,000 × 10^9^/L in a bleeding patient	Platelets 10 mL/kg
Antithrombin (AT)	>50–80%, if anticoagulation goals cannot be achieved by maximum UFH dose	Antithrombin concentrate Desired AT − Current AT × weight (kg)/1.4

Abbreviations: aPTT: activated partial thromboplastin time; RBC: red blood cells.

## Data Availability

Not applicable.

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
