# Peer review of "Extracorporeal Membrane Oxygenation (ECMO)-Associated Coagulopathy in Adults"

_diagnostics, 2023, doi:10.3390/diagnostics13233496_

Round 1

Reviewer 1 Report

Comments and Suggestions for Authors

I congratulate the authors to provide a comprehensive view on the topic of coagulopathy on ECMO. The topic is well researched and presented. However, I have few comments:

1. It would be nice if author could provide a diagram showing pathogenesis of caoagulopathy on ECMO as described in the text.

2. Please describe in the table 1 a  column of various values of the test.

3. Please  provide a table/algorithm for management of the coagulopathy. 

4. Please dedicate a heading on diffrerences in coagulopathy in VV vs VA ECMO and more so in Central Vs Peripheral VA ECMO. 

5. Please also contribute a section on Paediatric part.

Reviewer 2 Report

Comments and Suggestions for Authors

Dear authors,

my compliments on this work. 
The issue you are dealing with is very relevant to our current practice, as thrombotic/bleeding issue remain an unsolved problem when using mechanical circulatory support in general, and ECMO in particular, rendering the use of these devices often problematic and sometimes accounts for a limit in their use. The article you wrote presents us is valid in that it presents this issue and analyses our current strategies and tools. i find it a very valuable summary of existing data and a possible starting point for further researxh.

Comments on the Quality of English Language

Very good quality, a few minor adjustments (mainly with use of commas), need to be made.

Author Response

We thank the reviewer for the kind comments. We have made the recommended adjustments

Reviewer 3 Report

Comments and Suggestions for Authors

Frantzeskaki and collegues provided an excellent overview about EAC.

Author Response

We appreciate the kind comment. 

Round 2

Reviewer 1 Report

Comments and Suggestions for Authors

Very well conceptualised, researched and presented  on a relevant topic. Well done.